# Study on Summer Overheating of Residential Buildings in the Severe Cold Region of China in View of Climate Change

**Yang Yu** [1,2], **Yu Shao** [1,2], **Bolun Zhao** [1,2], **Jiahui Yu** [1,2], **Haibo Guo** [1,2,*] **and Yang Chen** [1,2,*]

1   School of Architecture, Harbin Institute of Technology, Harbin 150001, China
2   Key Laboratory of Cold Region Urban and Rural Human Settlement Environment Science and Technology, Ministry of Industry and Information Technology, Harbin 150001, China
*   Correspondence: guohb@hit.edu.cn (H.G.); chenyang1109@hit.edu.cn (Y.C.)

**Abstract:** Due to global warming, the overheating risk in the severe cold region of China has attracted attention, but so far, no studies have examined summer overheating in this region. This paper aims to reveal the overheating risk in recent and future climates in the severe cold region of China. An 18-storey residential building in the severe cold region of China was monitored from May to September 2021 to validate the simulation data of the indoor temperature. Weather files of the typical meteorological year (TMY) from 2007 to 2020, observations in 2021, and forecasts for the climate in different carbon emission scenarios (2030, 2060) were used to simulate the indoor temperature and assess the overheating risk. The results revealed the severity of the overheating risk; the overheating hours in the south-facing bedroom were recorded as 884 h (24.07%) with the TMY weather data and 1043 h (28.40%) in 2030 and 1719 h (46.81%) in 2060 under the RCP8.5 carbon emission scenario. Thus, the low carbon emissions policy may significantly alleviate overheating. Moreover, to cope with climate change, it is suggested that the Chinese local design standards should consider the summer overheating risk and make the necessary adjustments.

**Keywords:** overheating; thermal comfort; simulation; climate change; low carbon policy





## 1. Introduction

Due to economic and population growth, anthropogenic greenhouse gas emissions have reached unprecedented levels (at least in the past 800,000 years) [1]. According to the IPCC, the warming of the climate is incontrovertible, and many of the observed changes since the 1950s have been unprecedented in recent millennia [2]. Extremely hot outdoor conditions in the temperate climate have become more intense and frequent. Global warming is expected to lead to a serious increase in the frequency, amplitude, and duration of extreme high-temperature events [3]. The high indoor temperature will affect the health of residents and increase the risk of death [4]. The productivity and learning ability of residents will also be affected by the indoor temperature [5].

An overheating risk has also attracted attention in the severe cold region of China, where buildings usually have thermal insulation to reduce the thermal energy demand in winter. Since 2000, China's $CO_2$ emissions have more than tripled, and China has surpassed the United States as the world's largest annual emitter, accounting for about a quarter of the total global annual emissions [6]. Studies have shown that, since the middle of the 20th century, the rate of increase experienced by the surface temperature in China has been higher than the global average [7], and the average temperature of China in 2018 was about 1.2 °C higher than in 1901 [8]. From this perspective, the severely cold region of China is also facing the risk of summer overheating. However, at present, there is no uniform evaluation standard for summer overheating and also no definitive local building standard for thermal insulation in the severe cold region of China.

Furthermore, the continuous emission of greenhouse gases will lead to further warming of the climate, which may have a serious, universal, and irreversible impact on humans

and ecosystems. By the middle of the 21st century, however, the extent of climate change will be substantially affected by the emission scenario [1]. Most areas will experience high temperatures at higher frequencies and for longer durations. It is also predicted that a fatal heat wave of about 60 days will occur in the mid-latitude region every year, which will affect 48–74% of the global population by 2100 [9]. To mitigate climate change, some countries in the world propose $CO_2$ emission policies. China announced the goal of achieving a carbon peak by 2030 and carbon neutrality by 2060.

In this context, this paper aims to reveal the overheating risk in the severe cold region of China using weather data from different climatic conditions: TMY data from 2007 to 2020, observational data from 2021, and forecast data for 2030 and 2060 in different carbon emission scenarios. This study aims to fill in the lack of research on the overheating risk in the severe cold region of China and verify the overheating risk in more detailed weather data. Additionally, the paper uses empirical data to validate simulation data to increase simulation accuracy. Furthermore, this paper puts forward suggestions for improving the current policy standards in the severe cold region and discusses the impact of different carbon emission scenarios on indoor overheating.

## 2. Literature Review

### 2.1. Overheating Risk

More than 400 major cities in the world have records of urban overheating. In the summer of 2003, the Netherlands, France, Britain, and Germany recorded outdoor temperatures of 35 °C, 37 °C, 38.5 °C, and 40.2 °C, respectively [10]. Yannas et al. monitored and analyzed selected London residences. Their results showed that, due to the highly insulated enclosure structure specified by the regulations, the overheating risk was increasing even in winter [11]. Hamdy et al. conducted a simulation study on Dutch residences and found that old buildings with little or no mechanical ventilation were at risk of overheating [12]. Lee et al. simulated typical terraced houses in London under different climate scenarios and demonstrated that overheating occurred for long periods [13]. Pyrgou et al. simulated representative Italian residential buildings and found that the indoor temperature of insulated buildings was higher than that of traditional non-insulated buildings, and the cooling requirements in extremely hot periods were more than three times greater [14]. Figueroa-Lopez et al. simulated the passive buildings in northern Spain and found that residential buildings with low energy consumption also risked overheating in warm seasons [15]. Elsharkawy et al. simulated a tower block in London and found that if the indoor ventilation was poor, the insulation of the enclosure would become an important factor in determining the overheating risk [16]. Quinn et al. simulated 285 homes in New York City. Their results showed that many families would experience overheating risk during extremely high-temperature events [17]. Gilani et al. simulated typical office spaces in 14 cities representing major climate regions in Canada and the United States and found that inefficient use of windows would lead to overheating risk [18]. Su et al. recorded that the overheating risk in South Korean cities led to an increase in the building cooling demand [19]. Ngarambe et al. simulated the resident buildings in South Korea and found that, due to climate change, the overheating risk in urbanization areas was more serious than that in rural areas [20].

Recent studies using building-performance simulations have found that the risk of residential overheating in the severe cold region of China is increasing. Yang et al. found that the average indoor temperature of residential buildings reached 28 °C in Baotou [21]. Monna S found that the average indoor temperature of south-facing bedrooms without air conditioning at noon reached 33.2 °C in the monitored residential buildings of Dalian [22]. Bo, R. et al. simulated residential buildings in four cities in the severely cold and cold regions of China and revealed typical overheating [23]. They also found that an increase in the insulation thickness could aggravate summer overheating risk [24]. Wang et al. found that compared with traditional buildings, the overheating hours of high-performance buildings increased by 40.6% in Tianjin [25].

As evidence from the literature shows, most studies on overheating risk were concentrated in Europe, the United States, and Canada. Studies on the summer overheating risk of residential buildings in the severe cold region of China were limited. Moreover, the existing research mainly used simulations when discussing the overheating risk. The lack of monitored data was not sufficient to draw convincing conclusions.

### 2.2. Thermal Comfort Standards

The European standard 15251 (EN 15251), American Society of Heating, Refrigerating and Air-Conditioning Engineers standard 55 (ASHRAE-55), and the Chartered Institution of Building Services Engineers (CIBSE) standards are widely accepted international thermal comfort standards (Table 1). EN 15251 [26] stipulates three acceptable indoor temperature ranges. ASHRAE 55-2017 [27], an American standard, presents two acceptable indoor temperature ranges: within one, 90% satisfactory is expected; a second one requires 80% satisfaction. Based on EN 15251, CIBSE proposes a series of summer overheating guides for building environmental design. CIBSE Guide A [28] recommends 26 °C as an acceptable indoor temperature for natural ventilation in residential buildings. Unlike the former static standards, CIBSE TM59 is a dynamic standard that recommends free-running buildings to conform to Category II in EN 15251.

**Table 1.** Thermal comfort standards.

| Standard | Category | | Formula |
|---|---|---|---|
| EN15251 | I (PPD < 6%; $-0.2 <$ PMV $< +0.2$) | | $T_{max} = 0.31\,T_{od} + 17.8 \pm 2.5$ |
| | II (PPD < 10%; $-0.5 <$ PMV $< +0.5$) | | $T_{max} = 0.31\,T_{od} + 17.8 \pm 3.5$ |
| | III (PPD < 15%; $-0.7 <$ PMV $< +0.7$) | | $T_{max} = 0.31\,T_{od} + 17.8 \pm 4.2$ |
| ASHRAE Standard 55 | I $_{80\%\ satisfactory}$ (PPD $\leq$ 20%; $0.85 <$ PMV $< +0.85$) | | $T_{max} = 0.31\,T_{od} + 17.8 \pm 3.5$ |
| | II $_{90\%\ satisfactory}$ (PPD $\leq$ 10%; $-0.5 <$ PMV $< +0.5$) | | $T_{max} = 0.31\,T_{od} + 17.8 \pm 2.5$ |
| CIBSE | Guide A | Fixed | 26°C |
| | TM59 | Dynamic | $T_{max} = 0.33\,T_{rm} + 18.8 \pm 2$ |
| GB 50736 | Fixed | I (PPD < 10%, $-0.5 <$ PMV $< +0.5$) | 24–26 °C |
| | | II (PPD < 27%, $-1 <$ PMV $< +1$) | 26–28 °C |
| GBT50785 | Regions I, II | I $_{90\%\ satisfactory}$ (PPD < 10%, $-0.5 <$ PMV $< +0.5$) | $T_{max} = 0.77\,T_{od} + 12.04$ |
| | | II $_{75\%\ satisfactory}$ (PPD < 25%, $-1 <$ PMV $< +1$) | $T_{max} = 0.73\,T_{od} + 15.28$ |
| | Regions III, IV, V | I $_{90\%\ satisfactory}$ (PPD < 10%, $-0.5 <$ PMV $< +0.5$) | $T_{max} = 0.77\,T_{od} + 9.34$ |
| | | II $_{75\%\ satisfactory}$ (PPD < 25%, $-1 <$ PMV $< +1$) | $T_{max} = 0.73\,T_{od} + 12.72$ |

(Data source: EN 15251 [European], ASHRAE 55-2017 [American Society of Heating, Refrigerating and Air-Conditioning Engineers], CIBSE Guide A, CIBSE TM 59, GB 50736 [Heating, Ventilation and Air Conditioning of Civil Buildings], GBT 50785 [Evaluation Standard for Indoor Thermal Environment in Civil Buildings]). Regions I and II represent the severely cold and cold regions of China; regions III, IV, and V represent the other regions of China.

In China, there is no uniform evaluation standard for summer overheating, and the relevant local standards have not considered the problem of the summer overheating risk in the severely cold region. There are two thermal comfort building standards, namely the Design Code for the Heating, Ventilation, and Air Conditioning of Civil Buildings (GB 50736) and the Evaluation Standard for Indoor Thermal Environment in Civil Buildings (GBT 50785). GB 50736 [29] presents two acceptable indoor temperature ranges: the temperature of Class I has a suggested maintenance of between 24 °C and 26 °C; the temperature of Class II has a suggested maintenance of between 26 °C and 28 °C. GBT 50785 [30] also presents two acceptable indoor temperature ranges: 90% occupant acceptability and 75–90% acceptability. The maximum and minimum acceptable temperatures are set to 30 °C and 16 °C, respectively.

### 2.3. Climate Change and Low Carbon Policy

The global average surface temperature in the future will mainly be determined by the cumulative emissions of $CO_2$. Depending on socioeconomic development and climate policy, the projections of $CO_2$ emissions vary widely. In its fifth assessment report (AR5),

the IPCC identified new "benchmark emission scenarios", referred to as representative concentration pathways (RCPs), as shown in Figure 1. RCPs are used to make projections for four different 21st-century pathways of global greenhouse gas emissions: a strict mitigation scenario (RCP2.6), two intermediate scenarios (RCP4.5 and RCP6.0), and a scenario with high greenhouse gas emissions (RCP8.5). The baseline scenario, between RCP6.0 and RCP8.5, is the scenario without additional emission limitations. RCP2.6 represents an optimistic scenario in which the global temperature does not exceed the pre-industrial temperature by more than 2 °C. Compared with 1986–2005, the change in the global average surface temperature during 2016–2035 is predicted to be similar under four RCP scenarios and may be in the range of 0.3 °C–0.7 °C.

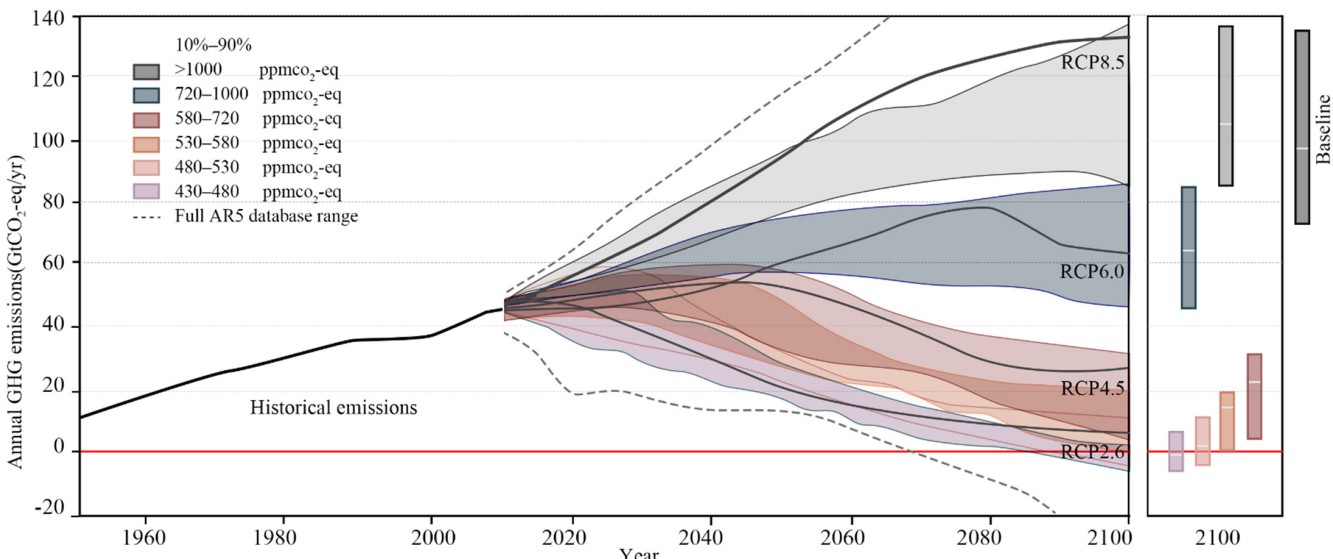

**Figure 1.** Global greenhouse gas emissions (gigatonne of $CO_2$-equivalent per year, GtCO₂-eq/yr) in baseline and mitigation scenarios for different long-term concentration levels (data source: Assessment Report 5 [2]).

To mitigate climate change, most countries in the world proposed $CO_2$ emission targets. Finland confirmed net zero emissions in 2035. Sweden, Austria, and Iceland, among other countries, aim to achieve net zero emissions in 2045. The European Union, the United Kingdom, Norway, Canada, Japan, and others set the goal for carbon neutralization in 2050. Some developing countries also plan to achieve carbon neutrality by 2050. China announced the goal of achieving a carbon peak by 2030 and carbon neutrality by 2060.

Some scholars have used predictions from weather models to analyze the impact of climate change on indoor thermal comfort [31,32]. Pagliano et al. argued that the discomfort duration in summer would be the main challenge in the future [33]. Yassaghi et al. simulated the meteorological data in 2020, 2050, and 2080 and observed that the cooling demand increased by 25.5–41.6% [34]. Zou et al. found that, due to the impact of climate change, the average number of overheating hours per month would double in 2041–2060 and increase by two to three times in 2081–2100 in Canada [35]. In China, Liu et al. found that the indoor discomfort rate in summer was expected to increase from 21.9% of the TMY to 36.0% under RCP4.5 and 50.4% under RCP8.5 by the end of this century [36]. Lei et al. found that, compared with the current climate scenario, the predicted average overheating hours in 2050 increased by 58–60% and 41–44% in Beijing and Shanghai [37]. Furthermore, the cooling energy consumption of buildings in Harbin and Beijing would also increase by 18.5% and 20.4% [38]. The research on the impact of climate change and carbon emission scenarios on summer overheating in the severe cold region of China is limited. If the current and future weather data are used for assessment, the overheating risk can be more comprehensively assessed.

## 3. Methods

The framework of the study is shown in Figure 2 and consists of four stages: measurement, validation, simulation, and evaluation. First, the indoor temperature of a real 18-storey residential building in Changchun was monitored by sensors from May to September 2021 to record building overheating time and validate the simulation data. Second, Pearson's R, regression equations, and RMSE were used to observe the fitting relationship between the monitored data and the simulation data to correct the simulation parameter settings with 2021 weather files. Third, the indoor temperature was simulated with the typical meteorological year (TMYx), weather files, and predicted weather files (2030, 2060). Fourth, the overheating risk was evaluated by CIBSE TM59 in three periods: the most recent TMY (2007–2020), 2021, and 2030, 2060.

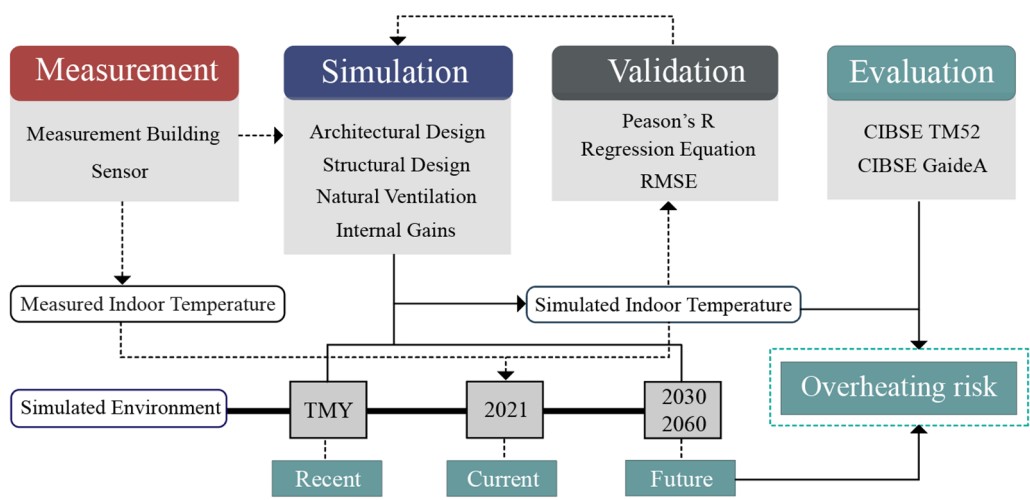

**Figure 2.** Framework of the study.

### 3.1. Measurement

An 18-storey concrete brick residential building in Changchun (Ib) was selected as a representative building in the severe cold region of China (Figure 3). The indoor temperature was monitored from 1 May to 30 September 2021 to correct the simulation parameter settings. The building is a typical residential building in the severe cold region, with four residential units on each floor. One living room facing south and two bedrooms (one facing south, the other north) in unit 10F were selected to be monitored from 1 May to 30 September 2022. All the rooms were predominantly naturally ventilated during the monitored period. The details of the residential building are shown in Table 2, and the room location is shown in Figure 4.

**Table 2.** Details of the residential building.

| Item | Values |
|---|---|
| Location | Changchun (125.19 E, 43.84 N) |
| Total floor layers | 18 F |
| Standard layer height | 3 m |
| Standard floor area | 461.8 m$^2$ |
| Room area | Living Room (Facing South):15.7 m$^2$ |
| | Bedroom (Facing South):13.5 m$^2$ |
| | Bedroom (Facing North):12.5 m$^2$ |

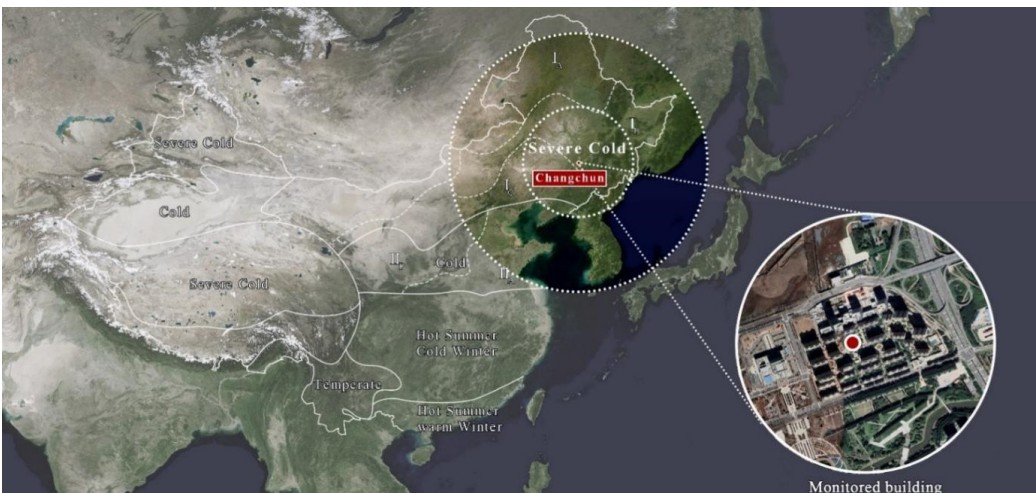

**Figure 3.** Chinese climate classification and the location of the monitored building.

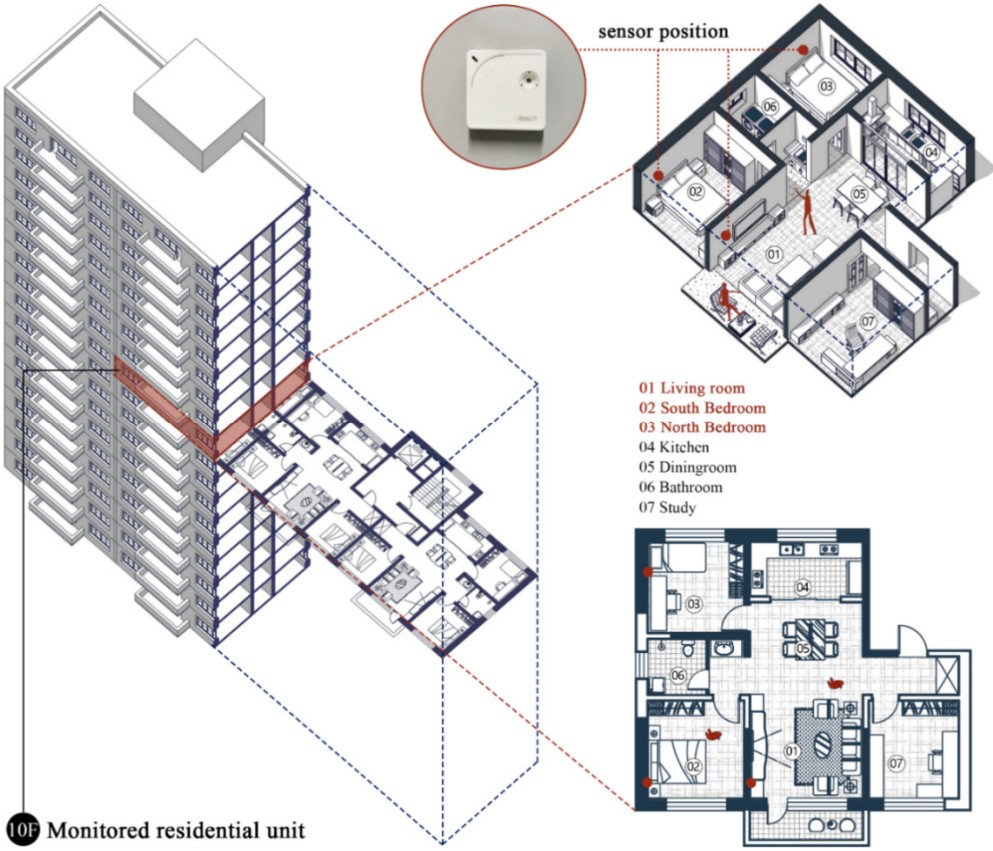

01 Living room
02 South Bedroom
03 North Bedroom
04 Kitchen
05 Diningroom
06 Bathroom
07 Study

**Figure 4.** Details of the architectural design.

The UbiBot-WS1 sensors were placed in each room to measure the indoor dry bulb temperature; the sensors were located on the wall 1.1 m above the floor to prevent the influence of direct sunlight. In each hour, 3672 measurements of the indoor temperature were collected in each room. The sensor installation positions and parameters are presented in Figure 4 and Table A1.

### 3.2. Simulation

The EnergyPlus software [39] was used to build a case model and simulate the indoor temperature of the selected rooms to evaluate the overheating hours in naturally ventilated

residential buildings. A simulation model consistent with the monitored residential building was established in the software. Three measured rooms were selected for simulation. The simulation parameters were set according to the actual situation. The weather data of 2007–2020, 2021, and 2030, 2060 were used in the simulation. Finally, EnergyPlus output the results of the zone air temperature (hourly) for the simulated rooms. The specific parameter settings are as follows.

### 3.2.1. Weather Data Settings

The simulations were conducted using four sets of weather data: TMY data from 2007 to 2020, representing the recent situation; observational data from 2021, representing the current weather; forecast data for 2030; and forecast data for 2060.

(1) The typical meteorological year in the Changchun TMY weather files was constructed over a 14-year period (from 2007 to 2020) and represented the recent climate scenario. The climate data are from the EnergyPlus website [39].

(2) As this study monitored the indoor temperature and validated the simulation data using the monitored data collected in 2021, those observational data were used as the current climate scenario.

(3) The predicted weather data between 2030 and 2060 with different emission scenarios generated by Meteonorm were used as the future climate scenario. This was based on three emission scenarios, namely RCP2.6 (low aggressive mitigation), RCP4.5 (moderately aggressive mitigation), and RCP8.5 (business as usual). Meteotest developed a tool named "Meteonorm" [40], which can be used for climate change research to generate future weather documents with a 10-year interval between 2010 and 2100.

The weather data for the four phases in May–September are shown in Figure 5, compared with TMYx data; the average summer outdoor temperature of 2030 and 2060 with RCP 8.5 increased by 1.45 °C and 3.16 °C.

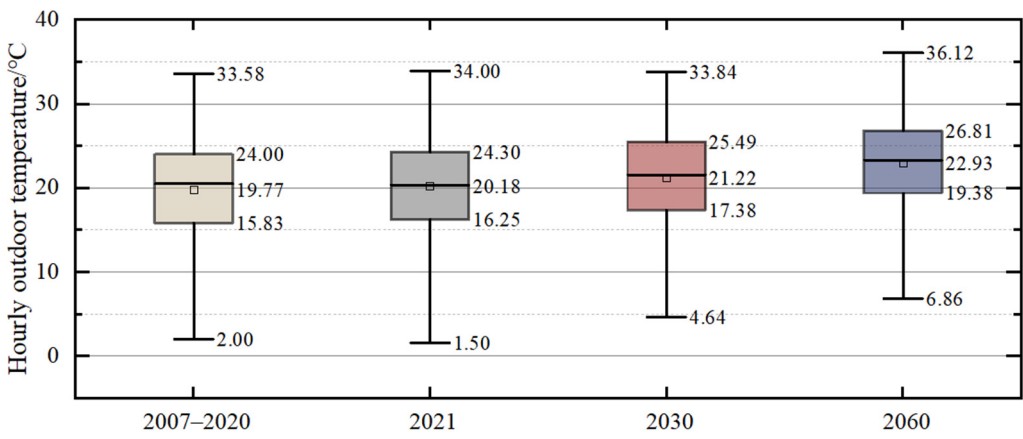

**Figure 5.** Outdoor temperature in different periods in Changchun.

### 3.2.2. Envelope Simulation Settings

The enclosure structure of the building consisted mainly of reinforced concrete, the internal partition wall was made of concrete bricks, and the floor was made of reinforced concrete. The U-value and g-value of the external wall, roof, ground, and window were defined by relevant national codes and standards. The design details are shown in Figure 6 and Table 3.

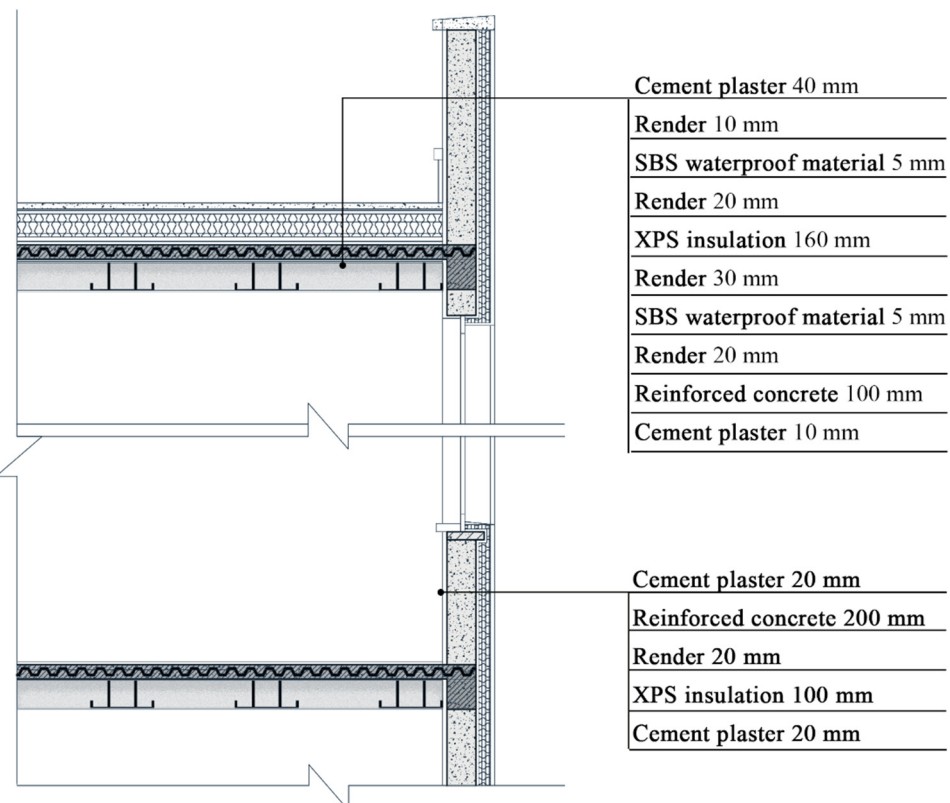

Cement plaster 40 mm
Render 10 mm
SBS waterproof material 5 mm
Render 20 mm
XPS insulation 160 mm
Render 30 mm
SBS waterproof material 5 mm
Render 20 mm
Reinforced concrete 100 mm
Cement plaster 10 mm

Cement plaster 20 mm
Reinforced concrete 200 mm
Render 20 mm
XPS insulation 100 mm
Cement plaster 20 mm

**Figure 6.** Envelope wall design for the residential building in Changchun.

**Table 3.** U-values of building envelope in the simulation settings.

| U-Values for Different Components (W/m² K) | | | | g-Value of the Windows |
|---|---|---|---|---|
| **External wall** | **Roof** | **Ground** | **Windows** | |
| 0.28 | 0.18 | 0.48 | 1.57 | 0.39 |

Data source: JGJ 26-2018 [Design Standard for Energy Efficiency of Residential Buildings in Severe Cold and Cold Zones] [41].

### 3.2.3. Ventilation and Internal Gains Simulation Settings

The residential building is considered a "natural ventilation" residential building as defined by CIBSE TM59. The internal gains, air exchange rates, and occupancy periods were set according to the General Specification for Building Energy Conservation and Renewable Energy Utilization (GB 55015-2021) and the surveys from the residents. The ventilation periods were set according to the outdoor temperature. For example, the windows were defined as open all day in July and August, which were seen as the hottest months in China. Additionally, the window opening time was decreased in May, June, and September when the weather was cooler. The heating and cooling were not considered in the simulation. Input parameter settings are shown in Table 4, and the detailed occupancy period settings are shown in Table A2.

**Table 4.** Ventilation and internal gain settings of the simulation building.

| Related Parameter | Input Parameter | Value | Period |
|---|---|---|---|
| Air exchanges | Infiltration | 0.3 ac/h | May–September (00:00–24:00) |
| | Natural ventilation | 3 ac/h | 1 May 1–5 June (08:00–20:00) 6–25 June (00:00–11:00, 14:00–24:00) 26 June—31 August (00:00–24:00) 1–17 September (07:00–18:00) 18–30 September (08:00–20:00) |
| Internal gains | People | 2/room | May–September |
| | Lighting | 5 W/m$^2$ | May–September |
| | Equipment | 3.8 W/m$^2$ | May–September |

Data source: GB 55015-2021 [the General specification for building energy conservation and renewable energy utilization] [42].

### 3.3. Validation

To ensure the accuracy of the simulation data, the indoor temperature was measured for data validation from May to September 2021. Pearson's R, the regression line, and RMSE were used to evaluate the accuracy of the simulation data [23,24]. Pearson's R was used to observe the fitting relationship between the measurement of indoor temperature (Meas) and the simulation of indoor temperature (Sim). When the value was close to 1, the simulation accuracy reached a high level. The k in the regression line and RMSE also indicate the relationship between Sim and Meas; when the value was close to 0, the simulation accuracy reached a higher level. Pearson's R, k, and RMSE can be calculated according to Equations (1)–(3), which can be shown by SPSS [43].

$$\text{Pearson's R} = \frac{\sum_{i=1}^{n}(\text{Meas}_i - \overline{\text{Meas}})(\text{Sim} - \overline{\text{Sim}})}{\left(\sqrt{\sum_{i=1}^{n}(\text{Meas}_i - \overline{\text{Meas}})^2}\right)\left(\sqrt{\sum_{i=1}^{n}(\text{Sim}_i - \overline{\text{Sim}})^2}\right)} \tag{1}$$

$$\text{Sim} = \text{k} \cdot \text{Meas} + \text{int} \tag{2}$$

$$\text{RMSE} = \frac{\sum_{i=1}^{n}(\text{Meas}_i - \text{Sim}_i)}{n} \tag{3}$$

In Equations (1) and (3), n represents the 3672 h from May to September, and i represents each hour during the study period. In Equation (2), k and int represent the slope and the intercept of the assumed linear equation.

To adapt to the simulated model with the future weather data, the ventilation period was appropriately adjusted according to the change in outdoor temperature in 2030 and 2060.

### 3.4. Standards for Evaluating Overheating

As the residential building was under natural ventilation, the CIBSE TM59 was selected as the assessment method. According to CIBSE TM59, Criteria (1) and (2) that follow must be passed for all relevant rooms.

According to CIBSE TM59 criteria (a) the number of hours in the living room and bedrooms during which $\Delta T$ is greater than or equal to one degree (°C) from May to September should not be more than 3% of the occupied hours. The occupied hours of the bedroom are 3672 h (24/7 from May to September), and 1989 h per year for the living room (13 h per day from May to September). The percentage of hours of exceedance over the whole study period (HE$_{(a)}$%) could be calculated according to Equations (4)–(6) below. $\Delta T$ is the difference between the operative temperature ($T_{op}$) and the maximum acceptable temperature ($T_{max}$), where T$_{max}$ can be calculated based on the running mean of the outdoor temperature ($T_{rm}$).

$$T_{rm} = (1 - \alpha)\left(T_{od-1} + \alpha T_{od-2} + \alpha^2 T_{od-3} \cdots\right) \tag{4}$$

$$T_{max} = 0.33T_{rm} + 21.8 \tag{5}$$

$$\Delta T = T_{op} - T_{max} \tag{6}$$

$$HE_{(a)}\% = \frac{\sum_{i=1}^{\text{occupied period}} wf_i, h_i}{\sum_{i=1}^{\text{occupied period}} h_i} \times 100\% \tag{7}$$

$$wf_i = \begin{cases} 1; & \Delta T \geq 0.5°\text{C} \\ 0; & \Delta T < 0.5°\text{C} \end{cases} \tag{8}$$

According to criteria (b), the indoor temperature in the bedroom from 10 pm to 7 am should not exceed 26 °C for more than 1% of the annual hours. $HE_{(b)}\%$ can be calculated as Equations (9) and (10).

$$HE_{(b)}\% = \frac{\sum_{i=1}^{\text{occupied period}} wf_i, h_i}{\sum_{i=1}^{\text{occupied period}} h_i} \times 100\% \tag{9}$$

$$wf_i = \begin{cases} 1; & T_{op} \geq T_{lim} \\ 0; & T_{op} < T_{lim} \end{cases} \tag{10}$$

## 4. Results

The results came from three aspects: the monitored data, simulation data, and validation data. Both the monitored data and the simulation data have proved that there was a serious overheating risk in the severe cold region of China. Validation data was well coupled with the monitored data and the simulation data, which confirmed the effectiveness of the simulation. The corresponding results are presented in chronological order as follows.

### 4.1. Monitored Results for 2021

According to the monitored data from 1 May to 30 September 2021, the monitored rooms in Changchun overheated. Figure 7 shows the indoor temperature of three monitored rooms. The rooms were under overheating conditions in July. The temperature range mainly fluctuated between 25–26 °C.

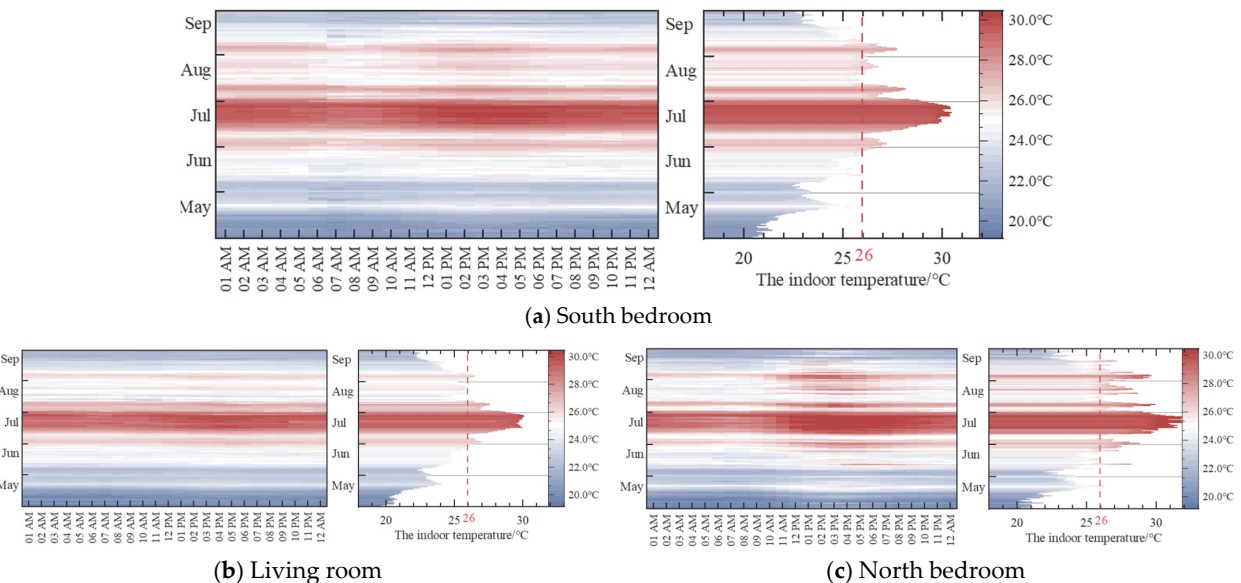

(**a**) South bedroom

(**b**) Living room

(**c**) North bedroom

**Figure 7.** Measured indoor temperature in temperature degree bands for three rooms from May to September in 2021.

According to CIBSE TM59, all the monitored rooms did not meet criteria (a) and (b). Table 5 shows the overheating hours of the three rooms based on CIBSE TM59. The temperature of all three rooms did not meet the thermal comfort standard. The overheating hours were recorded at 69 h (3.47% of the total) in the living room, 157 h (4.28%) in the south-facing bedroom, and 279 h (7.60%) in the north-facing bedroom. Moreover, the indoor temperatures that exceeded 26 °C during the sleeping hours were 420 h (12.79%) and 309 h (9.41%) for the south and north bedroom.

**Table 5.** Overheating hours of monitored rooms during summertime (May to September) 2021.

| Criteria (a) | | | | | | Criteria (b) | | | |
|---|---|---|---|---|---|---|---|---|---|
| Living Room | | Bedroom (South) | | Bedroom (North) | | Bedroom (South) | | Bedroom (North) | |
| Occupied Period ΔT ≥ 1 °C (8 a.m–9 p.m) | | Occupied Period ΔT ≥ 1 °C (24 h) | | | | Operative Temperature > 26 °C (10 p.m–7 a.m) | | | |
| HE/h | %HE/% | HE/h | %HE/% | HE/h | %HE/% | HE/h | %HE/% | HE/h | %HE/% |
| 69 | 3.47% | 157 | 4.28% | 279 | 7.60% | 420 | 12.79% | 309 | 9.41% |

### 4.2. Validation Results for 2021

To increase the accuracy of the simulation results, verification was carried out according to the measured data and the simulation data. By adjusting the ventilation parameters, the simulation data was gradually close to the measurement data. Figure 8 illustrates the regression lines of the three rooms. The k-values in the regression line of the three rooms were 1.11 in the living room, 1.08 in the south-facing bedroom, and 0.99 in the north-facing bedroom, respectively. Pearson's R were 0.82053 in the north-facing bedroom, 0.8586 in the south-facing bedroom, and 0.883 in the living room, respectively. While RMSE were 1.74388 °C, 1.85676 °C, and 1.93382 °C, respectively. These indicated a strong correlation between the simulated data and the measured data.

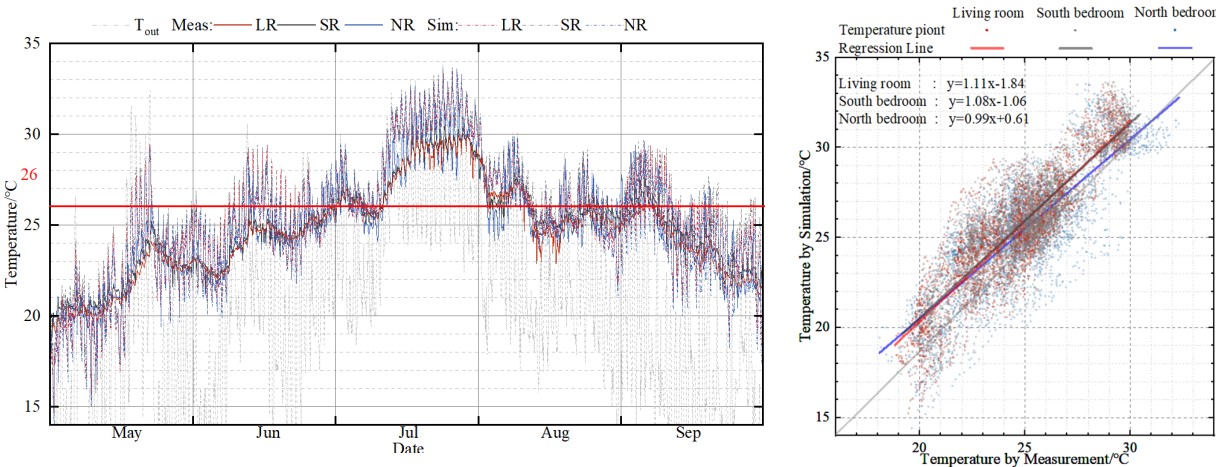

**Figure 8.** Validation data of three rooms in 2021.

### 4.3. Simulation Results for 2007–2020

After the model validation, the TMY weather file was applied to Energyplus in order to check the overheating risk during a 14-year period (2007–2020). According to the simulation results, all the rooms were overheated. Table 6 shows the overheating hours based on CIBSE TM59. The overheating hours were recorded for 529 h (26.60% of the total) in the living room, 742 h (20.21%) in the south-facing bedroom, and 361 h (9.83%) in the north-facing bedroom, respectively. Moreover, the indoor temperatures that exceeded 26 °C during the sleeping hours were 539 h (16.41%) and 406 h (12.36%) for the south and north bedroom. Bo, R. et al. simulated a residential building in Harbin, which was in the same climate

region as Changchun. The results showed that the indoor temperature exceeded 26 °C during the sleeping hours and was 12.4% of the south-facing bedroom [23], which is close to the simulation results in this paper. Figure 9 shows the indoor temperature and the overheating hours of the south-facing bedroom with TMY (2007–2020) weather data. The results indicate that the overheating periods were mainly in July, August, and September. The temperature range mainly fluctuated between 25 and 27 °C. Furthermore, the time that the indoor temperature exceeded 26 °C was also more than 50% of the simulation period.

**Table 6.** Overheating hours of simulated rooms based on CIBSE during summertime (May to September) 2007–2020 based on CIBSE TM59.

| | Criteria (a) | | | | | | Criteria (b) | | | |
| | Living Room | | Bedroom (South) | | Bedroom (North) | | Bedroom (South) | | Bedroom (North) | |
| | Occupied Period $\Delta T \geq 1$ °C (8 a.m–9 p.m) | | Occupied Period $\Delta T \geq 1$ °C (24 h) | | | | Operative Temperature >26 °C (10 p.m–7 a.m) | | | |
| | HE/h | %HE/% | HE/h | %HE/% | HE/h | %HE/% | HE/h | %HE/% | HE/h | %HE/% |
| Mode A | 529 | 26.60% | 742 | 20.21% | 361 | 9.83% | 539 | 16.41% | 406 | 12.36% |
| Mode B | 465 | 23.38% | 622 | 16.94% | 339 | 9.23% | 461 | 14.03% | 368 | 11.20% |
| Mode C | 411 | 20.66% | 524 | 14.27% | 286 | 7.79% | 404 | 12.30% | 320 | 9.74% |
| Mode D | 389 | 19.56% | 490 | 13.34% | 284 | 7.73% | 376 | 11.45% | 314 | 9.56% |

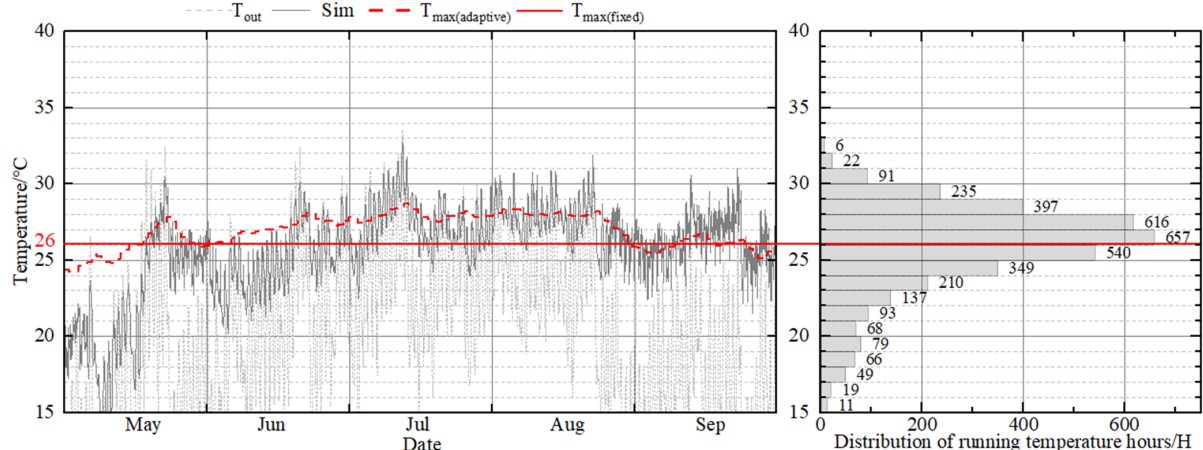

**Figure 9.** Indoor temperature of south-facing bedroom in temperature degree bands from May to September 2007–2020.

To verify the relationship between overheating and the ventilation mode, Table 6 also shows the simulation data for 2007–2020. Four ventilation modes (A, B, C, D) were simulated, including the original ventilation, increasing night ventilation, increasing ACH, and increasing both the ACH and night ventilation. The results show that the adjustment of ventilation could decrease the overheating hours significantly. Mode D had the best effect on reducing overheating hours.

### 4.4. Simulation Results for 2030 and 2060

After simulation with the current weather data, the overheating risk was assessed with the future weather data in different carbon emission scenarios to make the results more comprehensive. The results show that different degrees of overheating occurred in all scenarios. Table 7 shows the overheating hours based on CIBSE TM59 for the three simulated rooms under different carbon emission scenarios in 2030 and 2060. Take the south-facing bedroom as an example; under the RCP2.6 carbon emission scenarios, the overheating hours were 889 h (24.21%) in 2030 and 1131 h (30.80%) in 2060. The indoor temperatures that exceeded 26 °C during sleeping hours were 541 h (16.47%) and 696 h

(21.19%). Under the RCP4.5 carbon emission scenarios, the overheating hours were 1007 h (27.42%) in 2030 and 1437 h (39.13%) in 2060. The indoor temperatures that exceeded 26 °C during sleeping hours were 561 h (17.08%) and 839 h (25.54%). Under the RCP8.5 carbon emission scenarios, the overheating hours were 1043 h (28.40%) in 2030 and 1719 h (46.81%) in 2060. The indoor temperatures that exceeded 26 °C during sleeping hours were 625 h (19.03%) and 892 h (27.15%). Compared with 2030, the overheating hours gradually increased to 2060 under different carbon emission scenarios. Figure 10 shows that the indoor temperature gradually increased as the outdoor temperature increased in different periods, especially in September. The distribution of indoor temperature hours also changed significantly and was mainly distributed between 24 °C and 29 °C in 2030 and 25 °C and 30 °C in 2060. The number of high-temperature hours increased significantly over time.

**Table 7.** Overheating hours of simulated rooms based on CIBSE in different carbon emission scenarios during summertime (May to September) for 2030 and 2060 based on CIBSE TM59.

| Period | Carbon Emission Scenarios | Criteria (a) | | | | | | Criteria (b) | | | |
|---|---|---|---|---|---|---|---|---|---|---|---|
| | | Living Room | | Bedroom (South) | | Bedroom (North) | | Bedroom (South) | | Bedroom (North) | |
| | | Occupied Period $\Delta T \geq 1\,°C$ (8 a.m–9 p.m) | | Occupied Period $\Delta T \geq 1\,°C$ (24 h) | | | | Operative Temperature >26 °C (10 p.m–7 a.m) | | | |
| | | HE/h | %HE/% | HE/h | %HE/% | HE/h | %HE/% | HE/h | %HE/% | HE/h | %HE/% |
| 2030 | RCP2.6 | 683 | 34.34% | 889 | 24.21% | 665 | 18.11% | 541 | 16.47% | 474 | 14.43% |
| | RCP4.5 | 687 | 34.54% | 1007 | 27.42% | 777 | 21.16% | 561 | 17.08% | 505 | 15.37% |
| | RCP8.5 | 785 | 39.47% | 1043 | 28.40% | 800 | 21.79% | 625 | 19.03% | 559 | 17.02% |
| 2060 | RCP2.6 | 797 | 40.07% | 1131 | 30.80% | 808 | 22.00% | 696 | 21.19% | 622 | 18.93% |
| | RCP4.5 | 967 | 48.62% | 1437 | 39.13% | 1101 | 29.98% | 839 | 25.54% | 756 | 23.01% |
| | RCP8.5 | 1143 | 57.47% | 1719 | 46.81% | 1377 | 37.50% | 892 | 27.15% | 809 | 24.63% |

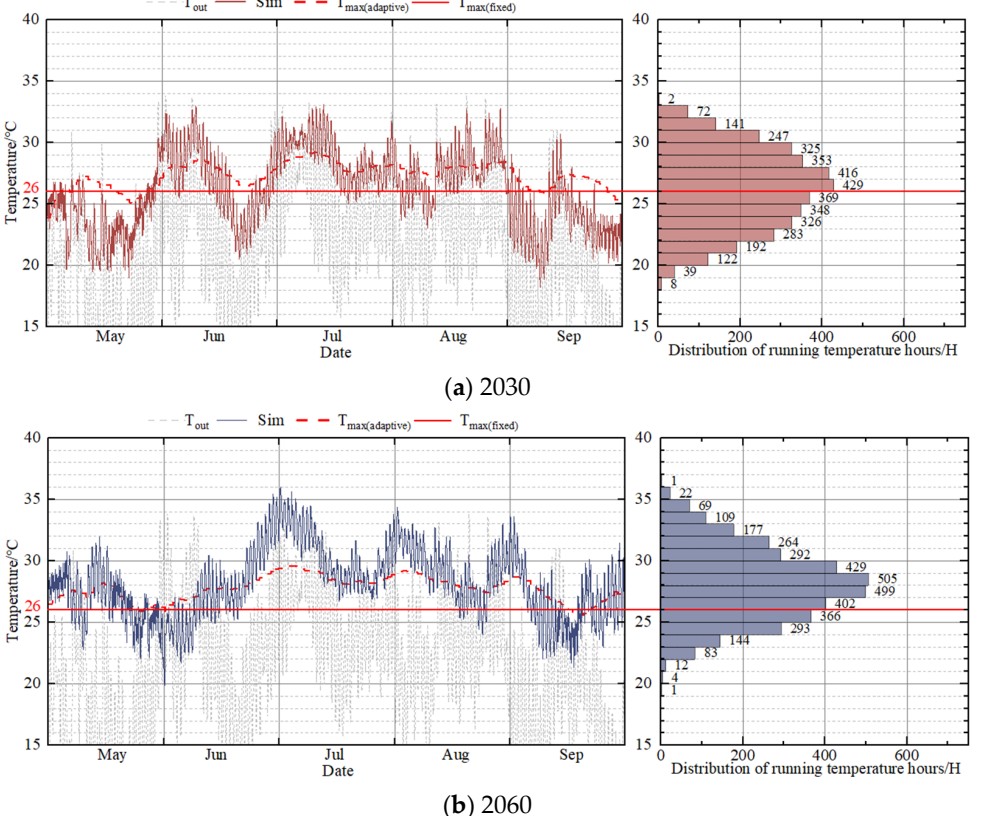

(**a**) 2030

(**b**) 2060

**Figure 10.** South-facing bedroom indoor temperature in temperature degree bands from May to September for 2030 and 2060.

## 5. Discussion

### 5.1. Overheating Risk and Policy Proposal in the Severe Cold Regions

Residences in the severe cold region have an overheating risk and may experience a more serious overheating risk with climate change in the future. Figure 11 shows the outdoor temperature and indoor temperature from May to September 2007–2020 and 2021, which is drawn from the resulting data in Figures 8 and 9. The average outdoor temperature in 2021 was 0.41 °C higher than the average temperature in 2007–2020, and the maximum temperature in 2021 was 0.42 °C higher than in 2007–2020. Furthermore, taking the south-facing bedroom as an example, compared with 2007–2020, the number of indoor high-temperature hours increased significantly in 2021; the maximum indoor temperature increased by 0.59 °C and the minimum indoor temperature increased by 1.07 °C. Furthermore, compared with 2021, the outdoor temperature for 2030 and 2060 with RCP 8.5 increased by 1.45 °C and 3.16 °C, and simulation overheating hours in the south-facing bedroom also increased by 194 h in 2030 and 870 h in 2060.

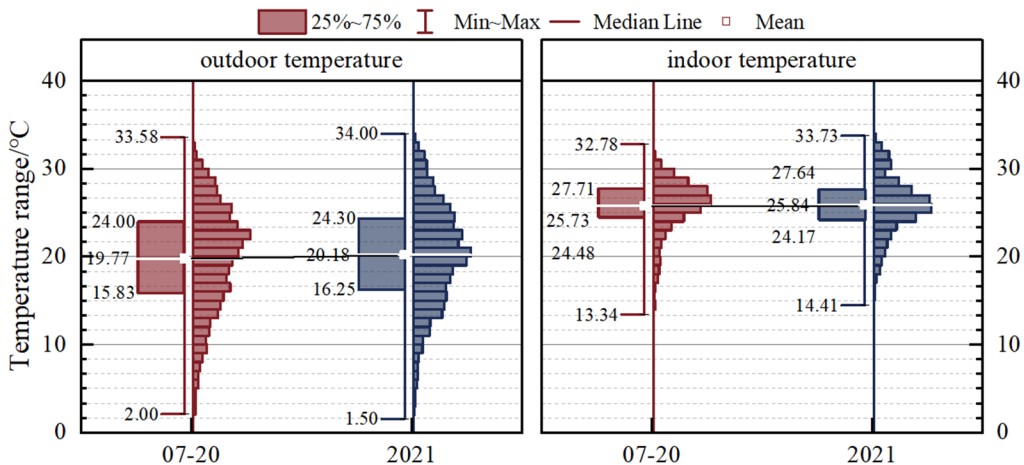

**Figure 11.** Outdoor temperature and indoor temperature from May to September for 2007–2020 and 2021.

Despite the current overheating risk and the trend of climate change, China has not yet implemented design standards or appropriate measures related to overheating in the severe cold region. The current national building codes and regional measures for thermal insulation in the summer are only recommended or specified in the II.B sub-region. The standards for the severely cold region mainly focus on thermal insulation measures in winter, which is relatively unfavorable to thermal insulation in the summer. The current standards should therefore be adjusted according to the overheating risk and climate change. The requirements for summer heat insulation in the severely cold region should be increased, and a balance should be achieved between winter heat insulation and summer heat insulation. For example, ventilation performance could be improved, and sunshade systems could be fitted to alleviate the overheating risk [44,45]. Fereidani et al. argued that the most effective passive cooling measures to cope with climate change in residential buildings are natural ventilation and thermal insulation [46].

This paper puts forward the following suggestions for the improvement of the standards. In terms of ventilation, night ventilation in summer should be considered during building design. The natural ventilation rate and permeability can be modified according to the actual use and relevant standards in hot summer and cold winter regions. In addition, the shading coefficient (SC) and solar heat gain coefficient (SHGC) should therefore be taken into account in this region (Table 8).

**Table 8.** Suggestions for revising China's summer thermal protection requirements and standards.

| Codes/Standards | | Content | Suggested Revisions |
| --- | --- | --- | --- |
| Code for Thermal Design of Civil Buildings (GB 50176-2016) | 4.3.2 | Heat insulation in summer should be considered in the building design of the II.B sub-region and not mentioned in IA, IB, IC of the severely cold region. | Heat insulation in summer should be considered in the design of buildings in the severely cold region in China. |
| | 6.3.1 | SHGC and SC shall should limited according to WWR in the II.B sub-region and not mentioned in IA, IB, IC of the severely cold region. | Sunshades should be considered in the severely cold region of China, and the limit values of SHGC and SC in summer should be given. |
| Design Standard for Energy Efficiency of Residential Buildings in Severely Cold and Cold Regions (JGJ26-2018) | 4.1.1 | Summer ventilation should be considered in the II.B sub-region and not mentioned in IA, IB, IC of the severely cold region. | Summer ventilation, such as night ventilation, should be considered in the severely cold region of China. |
| | 4.2.2 | SHGC should be limited according to Window to Wall Ratio (WWR) in the II.B sub-region and not mentioned in IA, IB, IC of the severely cold region. | Sunshades should be considered in the severely cold region of China, and the limit values of SHGC and SC in summer should be given. |
| | 4.2.4 | Sun shading should be considered in the II.B sub-region and not mentioned in IA, IB, IC of the severely cold region. | Sunshades should be considered in the severely cold region of China. |
| | 4.3.6 | The lower limit of infiltration ACH is 0.5 h$^{-1}$. | The ACH of infiltration can be increased according to the outdoor temperature. |

Data source: GB 50176-2016 [Code for Thermal Design of Civil Buildings], JGJ 26-2018 [Design Standard for Energy Efficiency of Residential Buildings in Severe Cold and Cold Regions].

The adjustment of ventilation modes is an effective way to alleviate summer overheating. Taking the simulated south-facing bedroom as an example, Figure 12 is drawn from the result data in Table 6, which shows the overheating hours in different ventilation modes. Compared with Mode A, using Mode B, Mode C, and Mode D could decrease the overheating hours by 120 h (3.27%), 218 h (5.94%), and 252 h (6.86%), respectively, based on CIBSE TM59. Additionally, the average indoor temperature decreased by 0.72 °C, 0.52 °C, and 1.12 °C, respectively. Heracleous studied the influence of ventilation in the educational buildings of Southern Europe with TMY weather data and found that night ventilation can reduce the overheating duration by 2% [47], which showed the effect of the ventilation adjustment. The adjustment of ventilation should be taken into account in policy formulation and architectural design to cope with future climate change in the severely cold region of China.

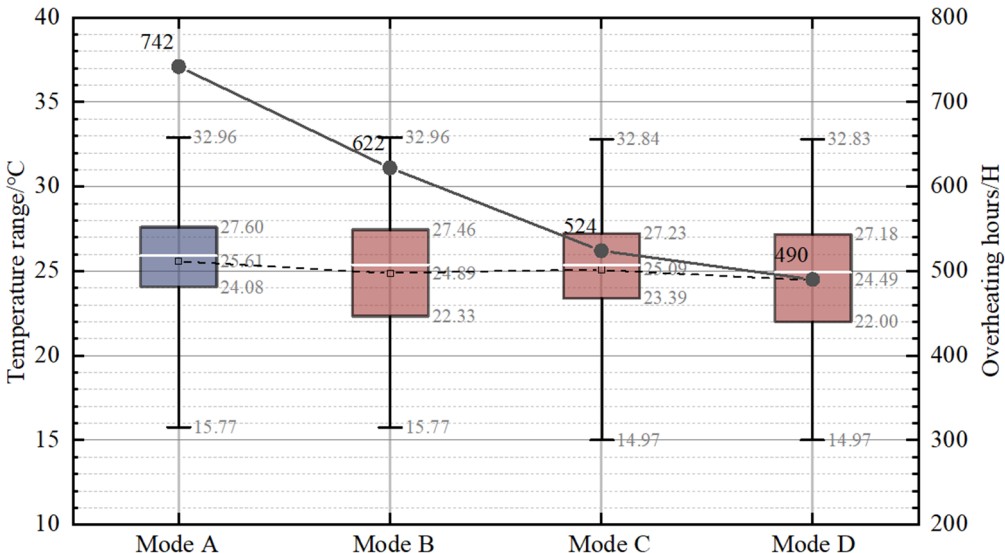

**Figure 12.** Influence of adjusting ventilation on indoor temperature of south facing bedroom from May to September 2007–2020.

*5.2. Influence of Low Carbon Policy Proposed by the Chinese Government on Indoor Temperature*

This paper used weather data from different climatic conditions and different carbon emission scenarios to discuss the influence of carbon emissions on indoor temperature. Different carbon emission scenarios could lead to different outdoor temperature conditions. Figure 13 shows the outdoor temperature under different carbon emission scenarios in 2030 and 2060, which is drawn from the result data in Table 7. Due to the short period, the outdoor temperatures in the three scenarios in 2030 were similar. The outdoor temperature in RCP2.6 was 0.19 °C lower than in RCP8.5. The difference was more obvious in 2060 when the outdoor temperature in RCP4.5 was 1.06 °C lower than in RCP8.5, and the outdoor temperature in RCP2.6 was 1.7 °C lower than in RCP8.5.

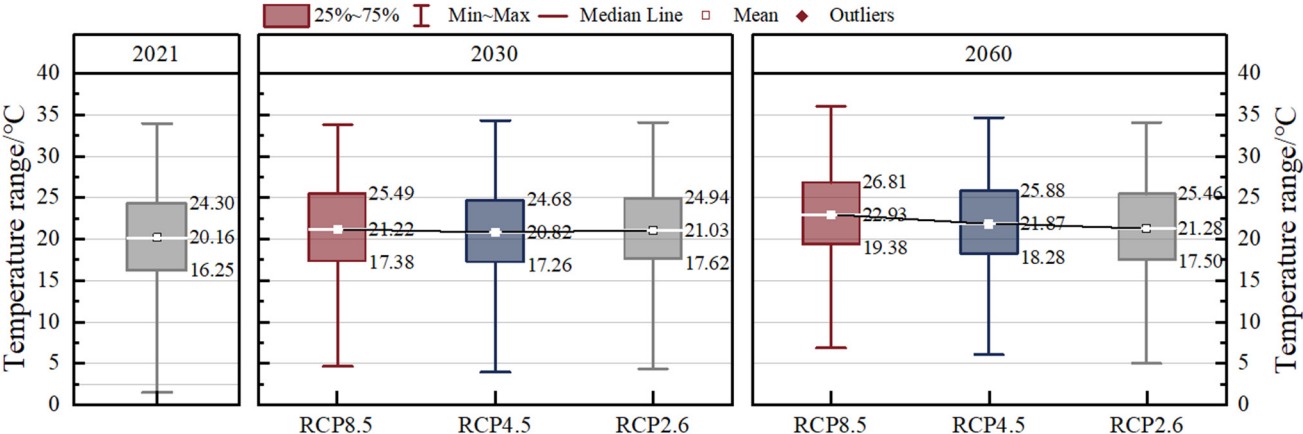

**Figure 13.** Outdoor temperature in different carbon emission scenarios in Changchun during summertime (May to September) in 2021, 2030, and 2060.

Low carbon emission scenarios could effectively reduce indoor overheating hours. Table 7 shows the overheating hours based on CIBSE TM59 of the three simulated rooms under different carbon emission scenarios between 2030 and 2060. Take the south-facing bedroom as an example; compared to the RCP8.5 carbon emission scenario, the overheating hours in 2030 decreased by 36 h (0.98%) in RCP4.5 and 154 h (4.19%) in RCP2.6. In 2060, the overheating hours decreased by 282 h (7.68%) in RCP4.5 and 588 h (16.01%) in RCP2.6. Figure 14 is drawn from the result data in Table 7, which shows the distribution of temperature hours in the south-facing bedroom during summertime in different carbon emission scenarios. In the low carbon emission scenario, the low-temperature hours increased while the high-temperature hours, especially above 28 °C, decreased significantly. Compared to the RCP8.5 carbon emission scenario, the high-temperature hours above 28 °C decreased by 105 h for RCP2.6 in 2030 and 771 h in 2060.

The dual carbon target strategy proposed by the Chinese government can greatly improve indoor thermal comfort and alleviate the summer overheating risk. In the Chinese low carbon policy, carbon emissions will continue to increase and reach a peak in 2030. Due to the short time period, the difference between high and low carbon emission scenarios in 2030 was not significant, which shows that, in view of the mid-term, the peak carbon emissions in 2030 will have little impact on summer overheating. China promised to achieve zero net carbon emissions in 2060. The difference between high and low carbon emission scenarios in 2060 was significant, which indicated that carbon emissions would have a significant impact on summer overheating.

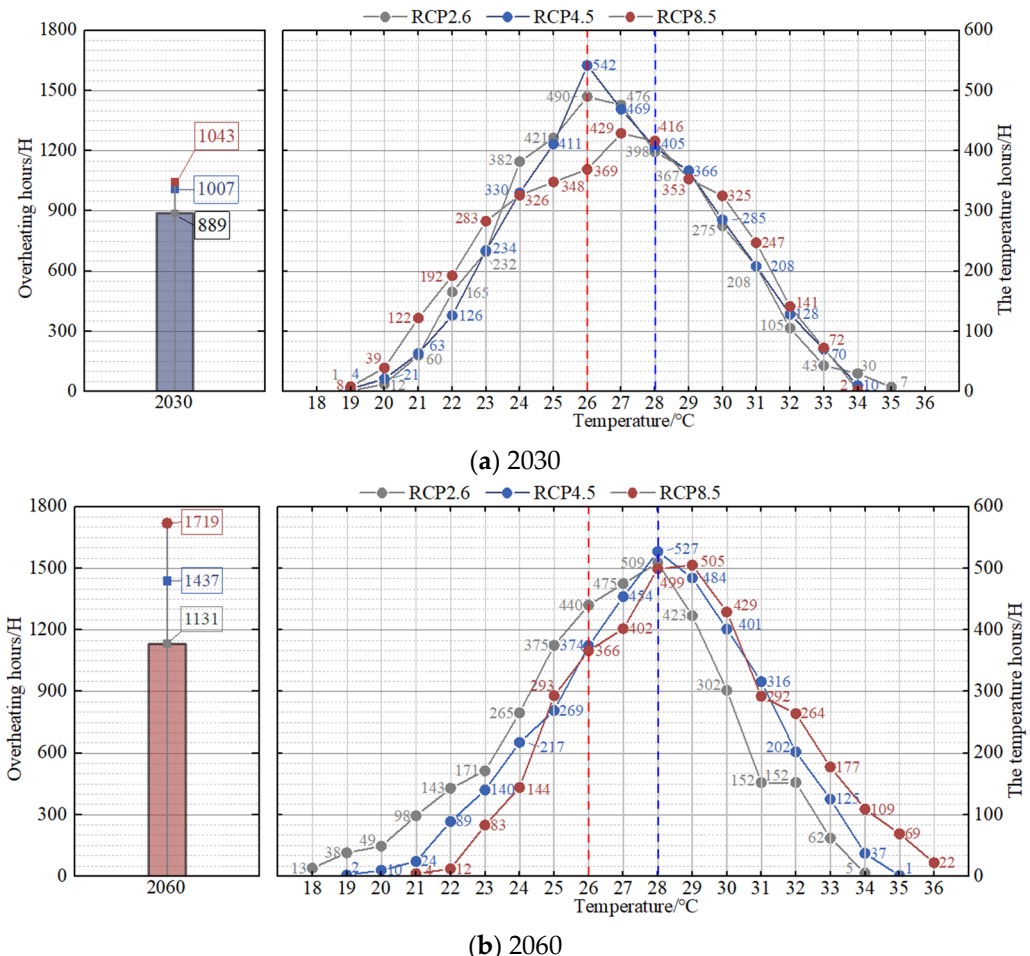

**Figure 14.** Overheating hours and the distribution of temperature hours in different carbon emission scenarios during summertime (May to September) for 2030 and 2060.

## 6. Conclusions

Due to global warming and the potential overheating risk in the severely cold region of China, this study aimed to reveal the overheating risk of residential buildings for recent and future climates in this region. The method of this paper combined the measurement with the simulation through validation to improve the authenticity of the simulation data. The relationships between climate change, carbon emission, and overheating risk were analyzed by simulation scenarios. Additionally, potential suggestions were put forward to improve the current policy standards. The main conclusions are as follows:

(1) The results show that residential buildings have a significant summer overheating risk in the severe cold region of China. In 2007–2020, the overheating hours were recorded at 559 h (28.10%) in the living room, 884 h (24.07%) in the south-facing bedroom, and 376 h (10.24%) in the north-facing bedroom. In the future, under the RCP8.5 carbon emission scenario, the overheating hours in the south-facing bedroom were 1043 h (28.40%) in 2030 and 1719 h (46.81%) in 2060.

(2) At present, there is no definitive local building standard for thermal insulation in the severe cold region of China. With the increased outdoor temperature and the potential overheating risk, it is suggested that the Chinese local design standards should consider the summer overheating risk and make necessary adjustments to climate change. The overheating hours of the south-facing bedroom were significantly reduced by 229 h (6.24%) through adjusting ventilation and ACH. Therefore, the adjustment of ventilation should be taken into account in policy formulation and architectural design in the severely cold region of China.

(3)　The low carbon emissions policy proposed by China might significantly alleviate the overheating risk. Compared to the RCP8.5 carbon emission scenario, the overheating hours of the south-facing bedroom decreased by 154 h (4.19%) in 2030 under the RCP2.6 carbon emission scenario, which showed that carbon emissions would have less influence on summer overheating in view of the mid-term. Compared to the RCP8.5 carbon emission scenario, the overheating hours were decreased by 588 h (16.01%) in 2060 under the RCP2.6 carbon emission scenario, which showed that the reduction in carbon emissions could greatly improve indoor thermal comfort and alleviate the summer overheating risk in view of the long-term.

**Author Contributions:** Conceptualization, Y.S. and Y.Y.; methodology, H.G.; Y.Y.; software, Y.Y. and J.Y.; validation, B.Z. and J.Y.; formal analysis, Y.S. and Y.C.; investigation, B.Z.; data curation, J.Y.; writing—original draft preparation, Y.Y.; writing—review and editing, H.G.; visualization, B.Z; supervision, Y.S. and Y.C.; project administration, H.G. and Y.C; funding/acquisition, H.G. All authors have read and agreed to the published version of the manuscript.

**Funding:** This research was funded by the National Natural Science Foundation of China, grant number 52078153.

**Institutional Review Board Statement:** Not applicable.

**Informed Consent Statement:** Not applicable.

**Data Availability Statement:** Not applicable.

**Conflicts of Interest:** The authors declare no conflict of interest.

## Appendix A

**Table A1.** Shows the sensor parameters in the monitored residential building. Sensor parameters.

| Parameter | Values |
|---|---|
| Dimension | $65 \times 65 \times 16.5$ mm |
| Acquisition parameters | Temperature |
| Network mode | WIFI (4.2 G) |
| Temperature range | $-20$–$60$ °C |
| Temperature accuracy | $\pm 0.3$ °C |

**Table A2.** Shows the occupancy, lighting, and equipment used in the simulated setting. Internal gain settings and occupancy periods of the simulation building.

| Related Parameter | Living Room (Facing South) | | | Bedroom (Facing South) | | | Bedroom (Facing North) | | |
|---|---|---|---|---|---|---|---|---|---|
| | People Occupancy (%) | Lighting Utilization (%) | People/Equipment Utilization (%) | People Occupancy (%) | Lighting Utilization (%) | People/Equipment Utilization (%) | People Occupancy (%) | Lighting Utilization (%) | People/Equipment Utilization (%) |
| 1:00 | 0 | 0 | 0 | 1 | 0 | 0 | 0.5 | 0 | 0 |
| 2:00 | 0 | 0 | 0 | 1 | 0 | 0 | 0.5 | 0 | 0 |
| 3:00 | 0 | 0 | 0 | 1 | 0 | 0 | 0.5 | 0 | 0 |
| 4:00 | 0 | 0 | 0 | 1 | 0 | 0 | 0.5 | 0 | 0 |
| 5:00 | 0 | 0 | 0 | 1 | 0 | 0 | 0.5 | 0 | 0 |
| 6:00 | 0 | 0 | 0 | 1 | 0 | 0 | 0.5 | 0 | 0 |
| 7:00 | 0.5 | 0.5 | 0.5 | 0.5 | 0.5 | 0.5 | 0.5 | 0.5 | 0.5 |
| 8:00 | 0.5 | 0.5 | 0.5 | 0.5 | 0.5 | 1 | 0.5 | 0.5 | 1 |
| 9:00 | 1 | 0 | 1 | 0 | 0 | 0 | 0 | 0 | 0 |
| 10:00 | 0.5 | 0 | 0.5 | 0 | 0 | 0 | 0 | 0 | 0 |
| 11:00 | 0.5 | 0 | 0.5 | 0 | 0 | 0 | 0 | 0 | 0 |
| 12:00 | 1 | 0 | 1 | 0 | 0 | 0 | 0 | 0 | 0 |
| 13:00 | 1 | 0 | 1 | 0 | 0 | 0 | 0 | 0 | 0 |
| 14:00 | 0.5 | 0 | 0.5 | 0 | 0 | 0 | 0 | 0 | 0 |
| 15:00 | 0.5 | 0 | 0.5 | 0 | 0 | 0 | 0 | 0 | 0 |
| 16:00 | 0.5 | 0 | 0.5 | 0 | 0 | 0 | 0 | 0 | 0 |
| 17:00 | 0.5 | 0 | 0.5 | 0 | 0 | 0 | 0 | 0 | 0 |
| 18:00 | 1 | 0 | 1 | 0 | 0 | 0 | 0 | 0 | 0 |
| 19:00 | 1 | 0 | 1 | 0 | 0 | 0 | 0 | 0 | 0 |
| 20:00 | 1 | 1 | 1 | 0 | 1 | 0.5 | 0 | 1 | 0.5 |
| 21:00 | 0.5 | 1 | 0.5 | 0.5 | 1 | 0.5 | 0.5 | 1 | 0.5 |
| 22:00 | 0 | 0.5 | 0 | 0.5 | 1 | 0.5 | 0.5 | 1 | 0.5 |
| 23:00 | 0 | 0 | 0 | 1 | 0 | 0 | 0.5 | 0 | 0 |
| 24:00 | 0 | 0 | 0 | 1 | 0 | 0 | 0.5 | 0 | 0 |

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
