# Peer review of "Study on Summer Overheating of Residential Buildings in the Severe Cold Region of China in View of Climate Change"

_buildings, doi:10.3390/buildings13010244_

Round 1
Reviewer 1 Report
This paper aims to reveal the overheating risk in recent and future climates in the severe cold region of China. I have some comments:
1. There paper has to be presice according to the template
2. Explain more detailed the results of the figure for example figure 9
Author Response
We appreciate the comments from the reviewer. Below are our responses to reviewer’s comments.
1 Comments: There paper has to be presice according to the template.
Reply: We have carefully checked and revised the paper format as your suggestions. Thanks for pointing this out. Please check all these through the paper.
2 Comments: Explain more detailed the results of the figure for example figure 9.
Reply: Thanks for pointing this out. We have added some explanations about the figure and table. More comprehensive explanations have been added to Figure 7, Table 5, and Figure 9.
Figure 7 shows the monitored data of the rooms in Changchun in 2021. The rooms were under overheating conditions in July. The temperature range mainly fluctuated between 25℃–26℃.
Please see lines 312-314.
Table 5 shows overheating hours of three rooms based on CIBSE TM59. The temperature of all three rooms did not meet the thermal comfort standard. The overheating hours were recorded 69 h (3.47% of the total) in the living room, 157 h (4.28%) in the south-facing bedroom, and 279 h (7.60%) in the north-facing bedroom. Moreover, the indoor temperatures that exceeded 26℃ during the sleeping hours were 420 h (12.79%) and 309 h (9.41%) of the south and north bedroom.
Please see lines 321-326.
Figure 9 shows the indoor temperature and the overheating hours of the south-facing bedroom with TMY (2007-2020) weather data. The results indicate that the overheating periods were mainly in July, August, and September. The temperature range mainly fluctuated between 25℃–27℃. What’s more, the time when the indoor temperature exceeds 26℃ was also more than 50% of the simulation period.
Please see lines 352-356.
Reviewer 2 Report
2.2 Simulation
More details for the EnergyPlus software are needed. Also give a reference.
Figure5: please write Figure 5
Table3: Please write Table 4
Figure6: Please write Figure 6
Please give a reference for Eqns 1-3
Figure7: please write Figure 7
Figure 8: Please change the first top figure with one with better analysis.
Author Response
We appreciate the comments from the reviewer. Below are our responses to reviewer’s comments.
1 Comments: More details for the EnergyPlus software are needed. Also give a reference.
Reply: Thanks for pointing this out. We have added references to EnergyPlus in the text and supplemented some details about its usage and output data.
The EnergyPlus software[37] was used to build a case model and simulate the indoor temperature of the selected rooms to evaluate the overheating hours in naturally ventilated residential buildings. A simulation model consistent with the monitored residential building was established in the software. Three measured rooms were selected for simulation. The simulation parameters were set according to the actual situation. The weather data of 2007-2020, 2021, and 2030,2060 were used in the simulation. Finally, EnergyPlus output the results of Zone Air Temperature (Hourly) of the simulated rooms. The specific parameter settings are as follows.
Please see lines 214-221.
2 Comments: Figure5: please write Figure 5, Figure6: Please write Figure 6, Table3: Please write Table 3, Figure7: please write Figure 7.
Reply: We have carefully checked and modified the number format as your suggestions.
Please see lines 248, 250, 251, 319.
3 Comments: Please give a reference for Eqns 1-3.
Reply: We have added references to Formula 1-3. Also, we have explained the meaning, evaluation criteria, and calculation methods of each formula.
To ensure the accuracy of the simulation data, the indoor temperature was measured for data validation from May to September in 2021. Pearson’s R, the regression line and RMSE are used to evaluate the accuracy of the simulation data[21, 40]. Pearson’s R is used to observe the fitting relationship between the measurement indoor temperature (Meas) and the simulation indoor temperature (Sim). When the value is close to 1, the simulation accuracy reaches a high level. The k in the regression line and RMSE also indicate the relationship between Sim and Meas, when the value is close to 0, the simulation accuracy reaches a higher level. Pearson’s R, k and RMSE can be calculated according to Equations (1)–(3), which can be shown by SPSS[41].
Please see lines 269-277.
4 Comments: Figure 8: Please change the first top figure with one with better analysis.
Reply: We have combined the three figures into one and used different colors to represent the proofreading data of different rooms. The first figure shows the measurement data and simulation data of the three rooms, while the second figure shows the fitting relationship between them. The closer the regression line is to y=x, the higher the fitting degree between the simulation data and the measurement data is. We also have added some corresponding explanations to the figure.
To increase the accuracy of the simulation results, verification was carried out according to the measured data and the simulation data. Through adjusting ventilation parameters, the simulation data was gradually close to the measurement data. Figure 8 illustrates the regression lines of three rooms. The k-values in the regression line of the three rooms were 1.11 in the living room, 1.08 in the south-facing bedroom, and 0.99 in the north-facing bedroom respectively. Pearson’s R were 0.82053 in the north-facing bedroom, 0.8586 in the south-facing bedroom, and 0.883 in the living room respectively. While RMSE were 1.74388℃, 1.85676℃, and 1.93382℃ respectively. These indicated a strong correlation between the simulated data and the measured data.
Please see lines 3329-339.
Reviewer 3 Report
The study submitted deals with the problem of residential buildings overheating in the severe cold region of China. The manuscript is well written and structured and the problem the authors describe is interesting. The methods used are sufficiently described. The results are appropriately discussed and the conclusions clearly described. The work is interesting, in terms of methodology results and can be easily transferred to other geographical areas.
Just a minor suggestion. In lines 430-432, to put emphasis on the fact that specifically in 2030, the difference between high and low carbon emission scenarios in 2030 is small, I suggest to add “in the mid-term”: “which shows that, in the mid-term, the carbon ….”
Author Response
We appreciate the comments from the reviewer. Below are our responses to reviewer’s comments.
1 Comments: In lines 430-432, to put emphasis on the fact that specifically in 2030, the difference between high and low carbon emission scenarios in 2030 is small, I suggest to add “in the mid-term”: “which shows that, in the mid-term, the carbon ….”
Reply: Thanks for your suggestion. We have added “which shows that, in view of the mid-term, the carbon emission reach the peak in 2030 has less influence on summer overheating”.
In Chines low carbon policy, carbon emissions will continue to increase and reach a peak in 2030. Due to the short time period, the difference between high and low carbon emission scenarios in 2030 is not significant, which shows that, in view of the mid-term, The peak carbon emissions in 2030 will have little impact on summer overheating.
Please see lines 467-470.
Reviewer 4 Report
This paper studied the summertime overheating of the residential buildings in China that were located in the cold region of the country to understand the impact of climate change on the severity of the overheating risks under the future climate scenarios. Although the topic was worth exploring, and valuable findings were gathered, the authors need to improve the manuscript as follows:
1. The Introduction section has a lack of logical flow. The authors provided the main findings of the other studies (simply a summary) without any critical argument, which will highlight what is the gap in the body of knowledge and how this research can help to fill this gap. The authors are recommended to separate the Introduction and the Literature review sections to be able to provide a background to the research problem in the Introduction section to indicate the aim of the research, while critically discuss the previous works on the topic in details in a standalone section. This will also help to point the knowledge gap in a more clear manner and show how this study can contribute to the current body of knowledge with evidence to emphasise its novelty.
2. In the Finding Section, it is recommended to provide references for each assumption deployed in the simulation settings such as U-values in Table 3 and ventilation and internal gain settings in Table 4.
3. The authors are suggested to highlight the differences between tables 7 and 9 since both include overheating hours of the simulated rooms with regards to the future climate in 2030 and 2060
4. Since the authors’ used indoor empirical data to calculate the summertime overheating for the current climate, whereas they used the outdoor climatic data to assess this risk, it is suggested to highlight how this process carried out in the simulation modelling and analysis to show the differences/similarities of the results.
5. The Discussion section includes the new findings with regards to the future climate scenarios. However, the Discussion section should not include any new findings/analysis. The authors are required to replace the section 4 into the Result section, while critically interpret and argue the main findings in the Discussion section in sequences, showing how these findings will fill the research gaps and highlighting how they move the field forward.
6. The Conclusion section is very short and shallow. It is suggested to expand the Conclusion section to provide an overview of the study, aim and the methods in one paragraph, while summarising the main findings in another paragraph to show how they will help to fill the gap.
7. The authors are recommended to proofread the manuscript and also make sure all the in-text citations are properly followed the references in the Reference list.
Author Response
We appreciate the comments from the reviewer. Below are our responses to reviewer’s comments.
1 Comments: The Introduction section has a lack of logical flow. The authors provided the main findings of the other studies (simply a summary) without any critical argument, which will highlight what is the gap in the body of knowledge and how this research can help to fill this gap. The authors are recommended to separate the Introduction and the Literature review sections to be able to provide a background to the research problem in the Introduction section to indicate the aim of the research, while critically discusses the previous works on the topic in details in a standalone section. This will also help to point out the knowledge gap in a more clear manner and show how this study can contribute to the current body of knowledge with evidence to emphasise its novelty.
Reply: Thanks for your suggestion. We have reorganized the structure of this article. Literature review section has been added after the introduction section. The literature review has been divided into three aspects: overheating phenomenon, overheating criteria, and future climate change. We have summarized each topic with the current research work.
Please see lines 25-179.
Also, in the result and discussion parts, we have added more research works and other examples to make comparisons with the results obtained in this paper to increase rationality and credibility.
The overheating hours based on CIBSE TM59 are shown in Table 6. According to the simulation results, all rooms were overheated apparently. The overheating hours were recorded 529 h (26.60% of the total) in the living room, 742 h (20.21%) in the south-facing bedroom, and 361 h (9.83%) in the north-facing bedroom respectively. Moreover, the indoor temperatures that exceeded 26℃ during the sleeping hours were 539 h (16.41%) and 406 h (12.36%) of the south and north bedroom. RuiBo et al. simulated a residential building in Harbin, which was in the same climate region as Changchun. The results showed the indoor temperature that exceeded 26℃ during the sleeping hours was 12.4% of the south-facing bedroom[20], which is close to the simulation results in this paper.
Please see lines 342-349.
Figure 12 shows the overheating hours in different ventilation modes. Compared with Mode A, using Mode B, Mode C, and ModeD could decrease the overheating hours by 120 h (3.27%), 218 h (5.94%), and 252 h (6.86%) respectively based on CIBSE TM59. And the average indoor temperature decreased by 0.72℃, 0.52℃, and 1.12℃ respectively. Heracleous studied the influence of ventilation in educational buildings of Southern Europe with TMY weather data and found night ventilation can reduce the overheating duration by 2%[45], which showed the effect of the ventilation adjustment.
Please see lines 434-440.
2 Comments: In the Finding Section, it is recommended to provide references for each assumption deployed in the simulation settings such as U-values in Table 3 and ventilation and internal gain settings in Table 4.
Reply: We have added references to the U-values, ventilation, and internal gain settings as your suggestion.
U-Values for Different Components(W/m2K) |
g-Value of the windows |
|||
External wall |
Roof |
Ground |
Windows |
|
0.28 |
0.18 |
0.48 |
1.57 |
0.39 |
Please see lines 251-253.
Related Parameter |
Input Parameter |
Value |
Period |
Air exchanges |
Infiltration |
0.3 ac/h |
May–September (00:00–24:00) |
Natural ventilation |
3 ac/h |
May.1- Jun.5(08:00-20:00) Jun.6-Jun.25(00:00-11:00, 14:00-24:00) Jun.26- Aug.31 (00:00-24:00) Sep.1-Sep.17 (07:00-18:00) Sep.18-Sep.30 (08:00-20:00) |
|
Internal gains |
People |
2 /room |
May–September |
Lighting |
5 W/m2 |
May–September |
|
Equipment |
3.8 W/m2 |
May–September |
Please see lines 256-259 and 265-267.
3 Comments: The authors are suggested to highlight the differences between tables 7 and 9 since both include overheating hours of the simulated rooms with regards to the future climate in 2030 and 2060
Reply: We have added more data to the result section, and combined Table 7 and Table 9 together.
Table 7 shows the overheating hours based on CIBSE TM59 of the three simulated rooms under different carbon emission scenarios in 2030 and 2060. Different degrees of overheating occurred in all scenarios. Take the south-facing bedroom as an example, under the RCP2.6 carbon emission scenarios, the overheating hours were 889h (24.21%) in 2030 and 1131h (30.80%) in 2060. The indoor temperatures that exceeded 26℃ during the sleeping hours were 541 h (16.47%) and 696 h (21.19%). Under the RCP4.5 carbon emission scenarios, the overheating hours were 1007h (27.42%) in 2030 and 1437h (39.13%) in 2060. The indoor temperatures that exceeded 26℃ during the sleeping hours were 561 h (17.08%) and 839 h (25.54%). Under the RCP8.5 carbon emission scenarios, the overheating hours were 1043h (28.40%) in 2030 and 1719h (46.81%) in 2060. The indoor temperatures that exceeded 26℃ during the sleeping hours were 625 h (19.03%) and 892 h (27.15%). Compared with 2030, the overheating hours gradually increased in 2060 under different carbon emission scenarios.
Period |
Carbon emission scenarios |
criteria (a) |
criteria (b) |
||||||||
Living room |
Bedroom (South) |
Bedroom (North) |
Bedroom (South) |
Bedroom (North) |
|||||||
Occupied Period ΔT≥1 ℃ (8a.m -9p.m) |
Occupied Period ΔT≥1 ℃ (24h) |
Operative Temperature >26 ℃ (10p.m-7a.m) |
|||||||||
HE/h |
%HE/% |
HE/h |
%HE/% |
HE/h |
%HE/% |
HE/h |
%HE/% |
HE/h |
%HE/% |
||
2030 |
RCP2.6 |
683 |
34.34% |
889 |
24.21% |
665 |
18.11% |
541 |
16.47% |
474 |
14.43% |
RCP4.5 |
687 |
34.54% |
1007 |
27.42% |
777 |
21.16% |
561 |
17.08% |
505 |
15.37% |
|
RCP8.5 |
785 |
39.47% |
1043 |
28.40% |
800 |
21.79% |
625 |
19.03% |
559 |
17.02% |
|
2060 |
RCP2.6 |
797 |
40.07% |
1131 |
30.80% |
808 |
22.00% |
696 |
21.19% |
622 |
18.93% |
RCP4.5 |
967 |
48.62% |
1437 |
39.13% |
1101 |
29.98% |
839 |
25.54% |
756 |
23.01% |
|
RCP8.5 |
1143 |
57.47% |
1719 |
46.81% |
1377 |
37.50% |
892 |
27.15% |
809 |
24.63% |
Please see lines 367-379 and 291-292.
4 Comments: Since the authors’ used indoor empirical data to calculate the summertime overheating for the current climate, whereas they used the outdoor climatic data to assess this risk, it is suggested to highlight how this process carried out in the simulation modelling and analysis to show the differences/similarities of the results.
Reply: The Chartered Institution of Building Services Engineers (CIBSE) has published various standards related to summer overheating. For naturally ventilated buildings, CIBSE TM59 provides two criteria (a, b) that combine static and adaptive methods for evaluating overheating. According to criteria (a), the number of hours in the living room and bedrooms during which ΔT is greater than or equal to one degree (℃) from May to September shall not be more than 3% of occupied hours. According to criteria (b), the indoor temperature in the bedroom from 10 pm to 7 am shall not exceed 26°C for more than 1% of annual hours. Criteria (1) and (2) must be passed for all relevant rooms, or the room will be defined as overheating.
Please see lines 288-309.
5 Comments: The Discussion section includes the new findings with regards to the future climate scenarios. However, the Discussion section should not include any new findings/analysis. The authors are required to replace the section 4 into the Result section, while critically interpret and argue the main findings in the Discussion section in sequences, showing how these findings will fill the research gaps and highlighting how they move the field forward.
Reply: According to your suggestion, we have transferred the data from the discussion part to the result part. We have combined Table 7 and Table 9 to describe the number of overheating hours in the future. Also, we have added Table 6 to the result to describe the impact of ventilation on summer overheating. Then, we have made specific discussion and analysis based on these results in the discussion part.
The content about Table 7 hase been displayed in Comments 3.
Please see lines 367-379 and 291-292.
The overheating hours based on CIBSE TM59 are shown in Table 6. According to the simulation results, all rooms were overheated apparently. The overheating hours were recorded 529 h (26.60% of the total) in the living room, 742 h (20.21%) in the south-facing bedroom, and 361 h (9.83%) in the north-facing bedroom respectively. Moreover, the indoor temperatures that exceeded 26℃ during the sleeping hours were 539 h (16.41%) and 406 h (12.36%) of the south and north bedroom. RuiBo et al. simulated a residential building in Harbin, which was in the same climate region as Changchun. The results showed the indoor temperature that exceeded 26℃ during the sleeping hours was 12.4% of the south-facing bedroom[41], which is close to the simulation results in this paper.
|
criteria (a) |
criteria (b) |
||||||||
|
Living room |
Bedroom (South) |
Bedroom (North) |
Bedroom (South) |
Bedroom (North) |
|||||
|
Occupied Period ΔT≥1 ℃ (8a.m -9p.m) |
Occupied Period ΔT≥1 ℃ (24h) |
Operative Temperature >26℃ (10p.m-7a.m) |
|||||||
|
HE/h |
%HE/% |
HE/h |
%HE/% |
HE/h |
%HE/% |
HE/h |
%HE/% |
HE/h |
%HE/% |
Mode A |
529 |
26.60% |
742 |
20.21% |
361 |
9.83% |
539 |
16.41% |
406 |
12.36% |
Mode B |
465 |
23.38% |
622 |
16.94% |
339 |
9.23% |
461 |
14.03% |
368 |
11.20% |
Mode C |
411 |
20.66% |
524 |
14.27% |
286 |
7.79% |
404 |
12.30% |
320 |
9.74% |
Mode D |
389 |
19.56% |
490 |
13.34% |
284 |
7.73% |
376 |
11.45% |
314 |
9.56% |
Please see lines 341-357.
6 Comments: The Conclusion section is very short and shallow. It is suggested to expand the Conclusion section to provide an overview of the study, aim and the methods in one paragraph, while summarising the main findings in another paragraph to show how they will help to fill the gap.
Reply: Thanks for your suggestion. the conclusion has been modified as follows.
Due to global warming and the potential overheating risk in the severe cold region of China, this study aims to reveal the overheating risk of residential buildings in recent and future climates in this region. The method of this paper combined the measurement with simulation through validation to improve the authenticity of the simulation data. The relationships between climate change, carbon emission, and overheating risk were analyzed by simulation scenarios. Also, potential suggestions were put forward to improve the current policy standards. The main conclusions are as follows:
(1) The results show the residential buildings have a significant summer overheating risk in the severe cold region of China. In 2007-2020, the overheating hours were recorded 559 h (28.10%) in the living room, 884 h (24.07%) in the south-facing bedroom, and 376 h (10.24%) in the north-facing bedroom. In the future under the RCP8.5 carbon emission scenario, the overheating hours in the south-facing bedroom were 1043h (28.40%) in 2030 and 1719h (46.81%) in 2060.
(2) At present, there is no definitive local building standard for thermal insulation in the severe cold region of China. With the increased outdoor temperature and the potential overheating risk, it is suggested that the Chinese local design standards should consider the summer overheating risk and make necessary adjustments to climate change. The overheating hours of the south-facing bedroom were significantly reduced by 229h (6.24%) by adjusting ventilation and ACH. So the adjustment of ventilation should be taken into account in policy formulation and architectural design in the severe cold region of China.
(3) The low carbon emissions policy proposed by China may significantly alleviate the overheating risk. Compared to the RCP8.5 carbon emission scenario, the overheating hours of the south-facing bedroom were decreased by 154h (4.19%) in 2030 under the RCP2.6 carbon emission scenario, which showed the carbon emission would have less influence on summer overheating in view of mid-term. Compared to the RCP8.5 carbon emission scenario, the overheating hours were decreased by 588h (16.01%) in 2060 under the RCP2.6 carbon emission scenario, which showed the reduction of carbon emission could greatly improve indoor thermal comfort and alleviate the summer overheating risk in view of long-term.
Please see lines 483-512.
7 Comments: The authors are recommended to proofread the manuscript and also make sure all the in-text citations are properly followed the references in the Reference list.
Reply: Thanks for your suggestion. We have carefully proofread and changed the citations in this paper.
Reviewer 5 Report
The paper fails to compare other studies from different regions with similar climates to the results obtained, which undermines the discussion section; therefore, consider to re-structure the literature review to a broader research representation, citing other examples that could enable a lengthy and interrelated discussion.
Author Response
We appreciate the comments from the reviewer. Below are our responses to reviewer’s comments.
1 Comments: The paper fails to compare other studies from different regions with similar climates to the results obtained, which undermines the discussion section; therefore, consider to re-structure the literature review to a broader research representation, citing other examples that could enable a lengthy and interrelated discussion.
Reply: Thanks for your suggestion. We have reorganized the structure of this article. Literature review section has been added after the introduction section. The literature review has been divided into three aspects: overheating phenomenon, overheating criteria, and future climate change. We have summarized each topic with the current research work.
Please see lines 25-179.
Also, in the result and discussion parts, we have added more research works and other examples to make comparisons with the results obtained in this paper to increase rationality and credibility.
The overheating hours based on CIBSE TM59 are shown in Table 6. According to the simulation results, all rooms were overheated apparently. The overheating hours were recorded 529 h (26.60% of the total) in the living room, 742 h (20.21%) in the south-facing bedroom, and 361 h (9.83%) in the north-facing bedroom respectively. Moreover, the indoor temperatures that exceeded 26℃ during the sleeping hours were 539 h (16.41%) and 406 h (12.36%) of the south and north bedroom. RuiBo et al. simulated a residential building in Harbin, which was in the same climate region as Changchun. The results showed the indoor temperature that exceeded 26℃ during the sleeping hours was 12.4% of the south-facing bedroom[20], which is close to the simulation results in this paper.
Please see lines 342-349.
Figure 12 shows the overheating hours in different ventilation modes. Compared with Mode A, using Mode B, Mode C, and ModeD could decrease the overheating hours by 120 h (3.27%), 218 h (5.94%), and 252 h (6.86%) respectively based on CIBSE TM59. And the average indoor temperature decreased by 0.72℃, 0.52℃, and 1.12℃ respectively. Heracleous studied the influence of ventilation in educational buildings of Southern Europe with TMY weather data and found night ventilation can reduce the overheating duration by 2%[45], which showed the effect of the ventilation adjustment.
Please see lines 434-440.
Round 2
Reviewer 4 Report
I appreciate the great effort that the authors have made to revise the manuscript and address the comments. The manuscript has been substantially improved. However, the authors are recommended to further improve the manuscript as follow to make the paper publishable:
1. I understand that there is limited research studies on the overheating risks in the severe cold region of China. However, the authors are required to introduce and describe a few studies that were carried out on overheating in the severe cold/cold regions of China or other countries with the similar climatic conditions and critically discuss their findings in the Literature Review section.
2. It is recommended to include an introductory paragraph into the Result section to attract the readers' attention and introduce them to the topic before presenting the data.
3. The Discussion section still includes new data that need to be explained in the Finding/ result section. If it is difficult for authors to separate the Result and Discussion section, they are suggested to combine the two section in a Result and discussion section, while critically argue the main Findings.
4. The authors are required to use a completely different line type/colour for each line graph in figures 8, 9 and 10 since it is difficult for readers to differentiate for example, Tout, Sim and Meas in the figures.
Author Response
We appreciate the comments from the reviewer. Below are our responses to reviewer’s comments.
1 Comments: I understand that there is limited research studies on the overheating risks in the severe cold region of China. However, the authors are required to introduce and describe a few studies that were carried out on overheating in the severe cold/cold regions of China or other countries with the similar climatic conditions and critically discuss their findings in the Literature Review section.
Reply: Thanks for your suggestion. We have added some studies on overheating in the severe cold and cold regions of China or other countries with similar climatic conditions (such as Europe, the United States, Canada, and South Korea). Also, we have critically discussed their findings in the Literature Review section.
Gilani et al. simulated typical office units in 14 cities representing major climate regions in Canada and the United States and found that inefficient use of windows would lead to overheating risk [18]. Su et al. recorded that the overheating risk in South Korean cities led to an increase in building cooling demand [19]. Ngarambe et al. simulated the resident buildings in South Korea and found due to climate change, the overheating risk in urbanization areas was more serious than that in rural areas [20]. Wang et al. found that compared with traditional buildings, the overheating hours of high-performance buildings increased by 40.6% in Tianjin [25].
As the evidence from literature shows, most studies on overheating risk were concentrated in Europe, the United States and Canada. Studies on summer overheating risk of residential buildings in the severe cold region of China were limited. Moreover, the existing research mainly used simulation when discussing the overheating risk. The lack of monitored data was not sufficient to draw convincing conclusions.
Please see lines 84-90 and 98-105.
2 Comments: It is recommended to include an introductory paragraph into the Result section to attract the readers' attention and introduce them to the topic before presenting the data.
Reply: We have added an introductory paragraph at the beginning of the results section. Also, we have added the theme of each part before presenting the data.
The results came from three aspects: monitored data, simulation data and validation data. Both the monitored data and the simulation data have proved that there was a serious overheating risk in the severe cold region of China. validation data has well coupled the monitored data with the simulation data, which confirmed the effectiveness of the simulation. The corresponding results were presented in chronological order as follows.
Please see lines 306-310.
According to the monitored data from May 1 to September 30 in 2021, the monitored rooms in Changchun have overheated.
Please see lines 312-313.
According to CIBSE TM59, all the monitored rooms did not meet the criterias (a) and (b).
Please see lines 322-323.
After the model validation, the TMY weather file was applied to Energyplus to check the overheating risk during a 14-years period (2007-2020). According to the simulation results, all rooms were overheated apparently.
Please see lines 343-345.
After the simulation with the current weather data, the overheating risk was assessed with the future weather data in different carbon emission scenarios to make the results more comprehensive. The results show different degrees of overheating occurred in all scenarios.
Please see lines 371-374.
3 Comments: The Discussion section still includes new data that need to be explained in the Finding/ result section. If it is difficult for authors to separate the Result and Discussion section, they are suggested to combine the two section in a Result and discussion section, while critically argue the main Findings.
Reply: Thanks for your suggestion, but the authors still think it is more reasonable to write the two parts separately. We have put some data in the discussion back to the results as suggested by the reviewer. The data of the existing figures under discussion were all from the results. The results were presented in chronological order, while the discussions were more focused on the reasonable proposal of building standards and the feasibility verification of dual-carbon policy based on the results.
We have added Table 6 to the results to describe the impact of ventilation on summer overheating. Also, Table 7 described the number of overheating hours in the future. Then, Figures 11, 12, 13, and 14 have made specific discussion and analysis based on these results in the discussion part.
Figure 11 shows the outdoor temperature and indoor temperature from May to September in 2007–2020 and 2021, which is drawn from the result data in Figure 8 and Figure 9.
Figure 12 is drawn from the result data in Table 6, which shows the overheating hours in different ventilation modes.
Figure 13 shows the outdoor temperature under different carbon emission scenarios in 2030 and 2060, which is drawn from the result data in Table 7.
Figure 14 is drawn from the result data in Table 7, which shows the distribution of temperature hours in the south-facing bedroom in different carbon emission scenarios.
Please see lines 401-403, 438-440, 455-456, and 470-472.
4 Comments: The authors are required to use a completely different line type/colour for each line graph in figures 8, 9 and 10 since it is difficult for readers to differentiate for example, Tout, Sim and Meas in the figures.
Reply: Thanks for your suggestion. We have replaced Figures 8, 9, and 10. The line graphs in each figure used completely different line types and colors.
